# Measurements of Plasma-Free Metanephrines by Immunoassay Versus Urinary Metanephrines and Catecholamines by Liquid Chromatography with Amperometric Detection for the Diagnosis of Pheochromocytoma/Paraganglioma

**DOI:** 10.3390/jcm9103108

**Published:** 2020-09-26

**Authors:** Wolfgang Raber, Hans Kotal, Rodrig Marculescu, Christian Scheuba, Martin B. Niederle, Alexandra Kautzky-Willer, Michael Krebs

**Affiliations:** 1Division of Endocrinology & Metabolism, Department of Medicine III, Medical University of Vienna, 1090 Vienna, Austria; h.kotal@gmx.at (H.K.); alexandra.kautzky-willer@meduniwien.ac.at (A.K.-W.); michael.krebs@meduniwien.ac.at (M.K.); 2Department of Laboratory Medicine, Medical University of Vienna, 1090 Vienna, Austria; rodrig.marculescu@meduniwien.ac.at; 3Department of Surgery, Medical University of Vienna, 1090 Vienna, Austria; christian.scheuba@meduniwien.ac.at; 4Department of Anesthesia, General Intensive Care and Pain Therapy, Medical University of Vienna, 1090 Vienna, Austria; martin.niederle@meduniwien.ac.at

**Keywords:** pheochromocytoma, diagnosis, plasma metanephrines, immunoassay, urinary metanephrines, urinary catecholamines

## Abstract

Studies conflict concerning the use of enzyme immunoassays (EIA) for plasma free metanephrines (P-MNs) vs. other methods for pheochromocytoma/paraganglioma (PPGL) diagnosis. We compared commercially available EIAs for P-MNs with high-pressure liquid chromatography (HPLC) for 24 h-urinary MNs (U-MNs) and -catecholamines (U-CATs). 943 (565 female, 378 male) patients (54 PPGL, 889 Non-PPGL) were studied. Simultaneous measurements of all parameters analyzed at the central lab of our university hospital was mandatory for inclusion. Sensitivity of P-MNs (94.4%) was similar to that of U-MNs (100%), and both were higher than of U-CATs (77.8%), specificity of P-MNs (100%) higher than of U-MNs (73.6%), and similar to U-CATs (99.8%). With the recently proposed downward adjusted ULN of P-MNs to correct for the reported negative bias of the EIAs sensitivity (98.1%) raised non-significantly, but specificity decreased significantly (94.8%). Areas under receiver-operating characteristic curves indicated comparable diagnostic performance of P-MNs (0.989) vs. U-MNs (0.995), both better than U-CATs (0.956). In summary, the EIAs to measure P-MNs performed similarly to U-MNs by HPLC, and both better than U-CATs by HPLC. The post-test probability of PPGL given a positive test result was best for P-MNs, and higher than for the other pairs of analytes. Downward corrections of ULN of P-MNs did not improve test performances.

## 1. Introduction

Pheochromocytomas (PHEO) and paragangliomas (PGL; together: PPGL) are rare catecholamine-producing tumors estimated to occur in about 2–8 out of 1 million people per year [1]. The prevalence in the hypertensive population is 0.1–0.5% [1,2,3], in an unselected hospital referral population 0.8–1.6% [4,5,6], and in patients with adrenal incidentalomas about 5% [7]. Those with a history of PPGL have a risk of recurrence as high as 16.5% [8], and 30–40% of PPGL may be associated with a hereditary disease [1,9], all scenarios typical of a large referral hospital and a specialized endocrine unit, respectively. The diagnosis may be challenging as about 50% of patients are asymptomatic [10] or suffer from only unspecific signs or symptoms [9,11]. On the other hand, up to 20% may develop life-threatening hemodynamic instability leading to pulmonary edema, malignant arrhythmias, myocardial infarction, stroke, or sudden death [12,13].

The 2014 Endocrine Society Clinical Practice Guideline with a level A quality of evidence by the GRADE system [14] recommends that initial biochemical testing should include the measurement of plasma-free or urinary fractionated metanephrines (MNs) with no preference of one over the other. The panel gives a grade 2 (weak) recommendation to use liquid chromatography (LC) with mass spectrometric (MS) or electrochemical detection (ED) rather than other lab methods [15]. Nonetheless, even in university hospitals, commercially available enzyme-linked immunoassays (EIAs) for the determination of plasma metanephrine (MN) and normetanephrine (NMN) are in widespread use due to their low price, ease of handling and—compared to chromatographic/MS methods—greater likelihood of being certified by ISO (International Organization for Standardization, Geneva, Switzerland) norms.

To date, no laboratory of the health care system of the Republic of Austria, in 2020 serving an estimated population of 8.9 millions (https://en.wikipedia.org/wiki/Austria), has been offering chromatographic/MS methods for quantification of plasma-free metanephrines for routine patient care, neither in private institutions nor in university hospitals. Capital costs of instrumentation, the level of operator expertise, and the need for in-house validation are higher than with EIA methods [16]. It is of note that in the European Union, until only recently, there has been no single officially CE-IVD (Conformité Européenne, In-Vitro Diagnostic Medical Device)-approved mass spectrometric or coulometric detector for the high-pressure (HP)LC to quantify plasma-free metanephrines—however, this approvement is mandatory for many labs due to regulatory legal requirements for the implementation of new methods. Based on the non-availability of these methods, we report our experience of *n* = 943 patients tested for PPGL in routine clinical care, in every patient and simultaneously using the three test methods routinely available (plasma-free metanephrines by EIA, 24-h urinary fractionated metanephrines, and 24-h urinary catecholamines, the latter two by HPLC with amperometric detection), compute which performs best for which metabolites to be analyzed, and compare the methods between each other. We think that the question of whether state-of-the-art care is possible without HPLC-MS/MS is of interest for analysts in all countries, where HPLC-MS/MS is not available. In 2015, the commercially available EIAs of Labor Diagnostika Nord, Nordhorn, Germany (LDN) have been reported to systematically underestimate plasma-free metanephrines contributing to a poor diagnostic sensitivity of only 74.1% in one series [17]. We thus wanted to know whether the correction algorithm, as proposed by Weismann et al. [17] to overcome their published negative bias of the EIAs suggested to have caused the low diagnostic sensitivity [17], is of clinical importance in a different population, or whether patient care would be possible without using these downward corrected upper limits of normal (ULN) of plasma-free metanephrines.

The study aims were: (1) Calculation of the diagnostic accuracy of the metabolites outlined in Table 1 as measured by the diagnostic sensitivity and specificity, by receiver–operating curve (ROC)-based analysis of the area under the curve (AUC) of single parameters, and of the pairs of measurements, and by using the posterior probability of PPGL given a positive and negative test result of the pairs of measurements, when upper limits of normal (ULN) stipulated by the manufacturers are applied; (2) assessment of the diagnostic accuracy of the EIAs following use of ULN to correct for the described negative bias as proposed recently [17], and comparison with the determination of the urinary metabolites.

## 2. Subjects and Methods

### 2.1. Subjects

The database consisted of 3373 patients from the Department of Laboratory Medicine of the Vienna General Hospital (in short, the lab). Clinical, biochemical, radiological, and histological data were obtained retrospectively. A mean of 1.4 (range: 1–5) samples for each plasma and 24 h-urinary measurements were obtained in patients with PPGL before surgery. Hereafter, only the first test results of the patients are considered for analysis.

The Vienna General Hospital generally serves as a tertiary referral center. Still, given the lack of insurance constraints in our country, many patients choose to attend the specialized university out-patient services for initial check-up of their complaints. Thus, for the purpose of this study, a low pre-test probability of PPGL can be reasoned. Two-thirds of the 3373 patients were seen either at the specialized endocrine out-patient unit or treated for various conditions as an in-patient. No attempt was made to differentiate between out-patients and those admitted to the hospital, nor to evaluate the medications taken by the patients at the time of biochemical evaluation. Only patients with all plasma and 24 h-urinary measurements, determined in-house and at the same time (with a tolerance of a maximum of one week apart), were included (Figure 1).

The indications for biochemical testing were assessed in 683 patients presenting with at least one positive measurement of any parameter: Hypertension resistant to ≥3 antihypertensive drugs or severe hypertension, or of young age (*n* = 346), the finding of an incidental adrenal mass (*n* = 120), symptoms suggestive of PPGL such as sweating, palpitations, or headache with or without paroxysmal hypertension (*n* = 101), suspected or confirmed genetic syndromes [1,9] associated with PPGL (*n* = 100), and a previous history of PPGL (*n* = 16). No effort was made to evaluate the indications for testing in the 260 patients presenting with negative measurements of all analytes at all times during follow-up (FU). Exclusion criteria are given in Figure 1.

*N* = 943 patients were thus included for analyses (Figure 1 and Table 2). Biochemical assessment was performed a mean of 7.6 times (range: 2–15) in patients assigned PPGL negative during the FU period of at least 2 years. Negative biochemical tests of all parameters at all times (from first presentation until ≥2 years of FU) were identified in *n* = 260 patients and were categorized as PPGL negative (PPGL-). Positive findings on at least one occasion of at least one analyte, (i) above the ULN stipulated by the manufacturer of the EIAs and the HPLC, respectively, or (ii) above the downward corrected ULN of P-MNs as proposed recently [17], were identified in *n* = 683 patients. The diagnosis of PPGL (PPGL+) was defined by histological criteria (*n* = 52), or by positive ^123^I-MIBG scan (*n* = 1) and positive F-DOPA PET-CT (*n* = 1) of two older patients presenting with grossly elevated concentrations of P-MNs, U-MNs, and U-CATs but refused surgery. *N* = 629 patients with at least one positive biochemical result were categorized PPGL- by best possible criteria (Table 2): By surgical exclusion of PPGL in *n* = 2 patients (one adrenal Cushing-syndrome and one adrenal adenoma), by normal thoraco-abdominal imaging (CT or MRI) and normal laboratory FU for at least 2 years (*n* = 158), by consistently normal biochemical findings together with absence or remission during a FU of ≥ 2 years of symptoms or signs suggestive of PPGL (*n* = 190), by an alternative diagnosis such as panic disorder or withdrawal from multiple substance abuse (*n* = 10), and by an uneventful clinical FU of ≥ 2 years including lack or remission of symptoms or signs suggestive of PPGL, and lack of perioperative complications during surgery for other conditions (*n* = 269).

### 2.2. Methods

All analyses were performed in an ISO 9001 certified and ISO 15189 accredited laboratory at the Department of Laboratory Medicine, Medical University of Vienna. All methods were approved for clinical diagnostic use according to the European Union In Vitro Diagnostics Directive 98/97/EG Conformité Européenne, In-Vitro Diagnostic Medical Device (CE IVD) labeled and were performed according to the manufacturers’ instructions for use. Analytical performance was constantly monitored by participation in international proficiency testing programs from INSTAND (Düsseldorf, Germany) and The Royal College of Pathologists of Australasia (RCPI, Surry Hills, NSW, Australia). In addition, the laboratory regularly exchanged samples and compared results with the Central Institute for Medical and Chemical Laboratory Diagnostics, Medical University of Innsbruck, Innsbruck, Austria according to Clinical and Laboratory Standards Institute [CLSI] Guideline QMS24. Plasma MNs were measured by the commercially available 2-MET Plasma ELISA Fast Track Assay kit of LDN [18] including two separate assays for P-MN and P-NMN, respectively. Since February 2013, all blood samples have been drawn according to standard operating procedures requiring a prior overnight fast and a supine resting position for at least 30 min. Before that date, about 70–80% of blood samples—corresponding to the percentage of patients from the endocrinology department—were obtained that way. No information was available of the other senders before February 2013. All blood samples were collected into K3EDTA tubes, plasma was separated by Fcentrifugation, and specimens were stored at −20 °C before analysis. Details are described elsewhere [16,17]. The intra-assay coefficients of variation (CV) ranged from 7.3 to 22 % for P-MN, and 7.2 to 7.8 % for P-NMN, the inter-assay CV from 12.0 to 21.1% for P-MN, and 11.7 to 13.0% for P-NMN. The lower limits of quantification (LLOQ) were 17 pg/mL (0.086 nmol/l) for MN, and 23 pg/mL (0.126 nmol/l) for NMN, respectively. The ULN were derived from fasting blood samples of 96 healthy subjects of the red cross Ljütensee near Hamburg, Germany (GER), and obtained in the sitting position (personal communication, Klemm R, Sales & Marketing Manager, LDN). The 24-h urinary measurements were by HPLC with amperometric detection using commercial kits of Chromsystems, Munich, Germany, as described previously [19]. The inter-assay CV ranged from 8.0 to 11.0% for NMN and MN, and from 5.0 to 8.0% for the CATs. Within-assay CV were <10% for both compounds. The LLOQ were 33, 5, 6, and 1 μg/24 h for NMN, MN, NA, and A, respectively.

Concentrations of test parameters are given as percent (%) of ULN to allow for easy comparisons between the methods. The ULN used for analyses are given in the Table 1.

### 2.3. Statistics

Sensitivity (defined as the fraction of patients with the disease identified as positive by the test), and specificity (fraction of those without the disease correctly identified as negative by the test) of the biochemical parameters were calculated for each analyte alone and as pairs (combined model). For the combined model, a positive finding was defined as a concentration above the ULN of at least one analyte, MN or NMN and A or NA, respectively. For a negative result, the concentrations of both analytes had to be below the respective ULN. Ninety-five percent confidence intervals (CI) of sensitivities and specificities were computed by the method of Wilson [20] and compared by McNemar’s test using a chi-square approximation [21].

To evaluate the diagnostic performance independent of ULN, ROC curves based on logistic regression analysis were created for each analyte as well as for the pairs, and 95% CI calculated by the method of Newcombe [22]. The ROC curves as a summary measure of the clinical utility of the tests show the relative changes in rates of true-positive (sensitivity plotted on the y-axis) and false-positive results (1-specificity plotted on the x-axis) for the diagnosis of PPGL as a function of the respective ULN. To test the null hypothesis of equal diagnostic performance of EIA and HPLC, the method of Hanley and McNeil [23] was applied. When the critical value Z of the standard normal distribution exceeded ±1.96 of the two-sided significance of 0.05, the null hypothesis of equal performance was rejected.

To determine the probability of a patient suffering from PPGL given a positive and a negative test result, Fagan´s nomogram for Bayes theorem based on the pre-test probability (numbers of PPGL divided by the total study population) and the diagnostic sensitivity and specificity was used [24,25]. Post-test probabilities (post-TP) with 95% CI were assessed with the diagnostic test calculator software version 201004210, free software available under the Clarified Artistic License [26]. All other computations were performed using GraphPad Prism version 8.0.0 for Mac, GraphPad Software, San Diego, CA, USA, www.graphpad.com. Means and medians were compared by the t-test and the Mann-Whitney U test (both two-sided), respectively, as appropriate. For the comparison of three or more means, one-way ANOVA and Tukey´s multiple comparisons test was used, for non-parametric comparisons of three or more medians the Kruskal–Wallis and Dunn´s multiple comparisons test. *P*-values < 0.05 were considered significant.

The study has been approved by the ethics committee of the Medical University of Vienna (No. 1089/2016).

## 3. Results

### 3.1. Baseline Data

Age and sex were comparable between PPGL+ and PPGL− patients (*p* = n.s.). A genetic disease was identified in 14 of the 54 patients with PPGL (seven with multiple endocrine neoplasia type 2, two with von Hippel-Lindau disease, two with neurofibromatosis, two with succinate dehydrogenase mutation B, and one with PGL-3 locus mutation). These patients were younger (*p* = 0.001) than those with sporadic disease (37 ± 12 vs. 53 ± 14 years), but concentrations of biochemical parameters were comparable (*p* = n.s.) to patients with sporadic tumors. There were 5 patients with PGL (2 of them with metastases), 2 patients with both PGL and PHEO, and 49 PHEO among the 54 patients with confirmed PPGL (Table 3). As to the possible distinctive biochemical profile of patients with adrenal (PHEO) versus extra-adrenal (PGL) tumor, parameters were analyzed separately for both entities. The latter displayed higher (*p* = 0.04 and 0.03, respectively) U-NMN and U-NA, but similar concentrations of the other parameters (Appendix A). The small number (*n* = 5) of PGL precludes further evaluation as to the accuracy of the diagnostic tests in PGL vs. PHEO.

### 3.2. Sensitivity and Specificity

#### 3.2.1. Sensitivity and Specificity of Single Parameters

U-NMN detected all 54 PPGL and provided higher (*p* < 0.05) sensitivity than all other single parameters, except for P-NMN (51 of 54 PPGL positive) with the corrected ULN as proposed by others [17], U-A and U-NA the lowest. On the other hand, P-NMN was most specific (4 of 889 individuals without PPGL were false positive), more (*p* < 0.05) than all other single metabolites except for U-A (six false positives). U-NMN (472 false positives) performed worst regarding specificity. Figure 2 shows the sensitivity and specificity of the analytes according to the ULN given in Table 1. Comparisons of sensitivities and specificities including 95% CI are given in Appendix A.

#### 3.2.2. Sensitivity and Specificity of Parameters Combined

Clinical decisions are seldom based on a single parameter, so sensitivities and specificities were computed of combined models (pairs) as well. The sensitivity (Figure 3, below the dashed line) of the pair of P-MN and P-NMN (51 of 54 PPGL detected) was higher (*p* ≤ 0.02) than that of the pair of U-A and U-NA (42 of 54 PPGL detected), but not (*p* = n.s.) vs. the combined U-MN and U-NMN (all 54 PPGL detected). Specificity (Figure 3, above the dashed line) was highest with the combined P-MN and P-NMN (no false positives of 889 patients without PPGL), and higher (*p* < 0.001) than that of the pair of 24 h-urinary metanephrines (236 false positive) and of plasma-free metanephrines (46 false positive), when using the corrected ULN as proposed by others [17].

As depicted in Figure 3, using this downward corrected ULN of plasma-free metanephrines as suggested by Weismann et al. [17] resulted in a statistically non-significant increased sensitivity versus no corrections of ULN (one vs. three false negative), at the cost of a significant decrease in specificity (0 vs. 46 false positive).

False positive results of U-MN and U-NMN showed a broader range (*p* < 0.05) up to almost 5-fold the ULN when compared to the range of false positive concentrations (up to 3.3-fold the ULN) of P-MN and P-NMN with the corrected ULN [17]. Comparisons of sensitivities and specificities including 95% CI are summarized in Table 4.

### 3.3. ROC Curves and Comparison of AUCs

The significance level *p* (area = 0.5) was equal to 0.0001 for all analytes alone and in pairs, indicating that all metabolites, determined by either method, were able to correctly classify patients with and without PPGL.

When single parameters were considered, P-NMN provided the largest AUC (0.978, 95% CI: 0.954–1.000) and was superior (*p* = 0.02, 0.004, and <0.0001, respectively) to U-MN (0.908, 95% CI: 0.856–0.960), to U-A (0.879, 95% CI: 0.817–0.940), and to U-NA (0.869, 95% CI: 0.812–0.927), respectively, but not better (*p* = n.s.) than U-NMN (0.960, 95% CI: 0.934–0.986). The AUC of P-MN (0.882, 95% CI: 0.824–0.940) was smaller (*p* = 0.02) than that of U-NMN, but not different (*p* = n.s.) to the other 24 h-urinary analytes (Figure 4, Table 5).

When the combined models were considered, both the AUC of the pairs of P-MN and P-NMN (0.989, 95% CI: 0.972–1.000), and of U-MN and U-NMN (0.995, 95% CI: 0.980–1.000), were larger (*p* < 0.05) than that of the pair of U-A and U-NA (0.956, 95% CI: 0.930–0.975), but not different (*p* = n.s.) when compared to each other (Figure 4, Table 5).

### 3.4. Post-Test Probability of PPGL

Given the 5.7% pre-test probability of PPGL in our study population (the 1.6% pre-TP based on the database of lab results of 3373 patients provided by the lab is not considered), the post-TP of a patient suffering from PPGL given a positive and negative test result of the pairs of analytes were 100% and 0% using P-MN and P-NMN together, 53% and 0% with the combined P-MN and P-NMN using the corrected ULN [17], 18% and 0% for the combined U-MN and U-NMN, and 96% and 1% for the combination of U-A and U-NA, respectively.

When analyzed with the higher ULN stipulated by the manufacturer of the 24 h-urinary metanephrines after February 2019, post-TP of PPGL given a positive test result increased to 76%, while that of a negative result remained the same. More details including summaries of sensitivities and specificities as well as calculations of post-TP of positive and negative test results based on the data of other studies are given in the Appendix A.

## 4. Discussion

To the best of our knowledge, this is the largest study comparing the diagnostic efficacy of an immunometric method for the measurement of plasma-free metanephrines (P-MNs) to that of other methods in the biochemical diagnosis of PPGL, and the only series that reports superiority of P-MNs over another method (24-h catecholamines, U-CATs). LC-MS/MS for the measurement of P-MNs or 24-h urinary metanephrines (U-MNs) have been recommended as one of the first-line tests in the biochemical diagnosis of PPGL [15]. Their availability is likely to be restricted to centers that have developed and optimized their in-house analyses, however. As outlined in the introduction, there is no lab offering a LC-MS/MS method for analysis of P-MNs in Austria. The EIAs of LDN on the other hand, developed in 2004 [27] and refined from one version to the next, are certified by ISO 9001 and ISO 13485, respectively, and satisfy the economic and medico-legal needs of public state hospitals such as the Vienna General Hospital, but still take more than 24 h to complete, probably limiting the introduction into routine use in smaller district hospitals. It may therefore be of great interest to all analysts in countries, where LC-MS/MS is not available, to know how the EIA for quantification of P-MNs perform compared to urinary measures.

Our study provides evidence of equal diagnostic performance as quantified by the AUC of ROC curves of a commercially available EIA kit for the measurement of P-MNs, when compared to a HPLC for U-MNs, and that both methods perform better than the determination of 24 h-urinary catecholamines (U-CATs) by HPLC. The use of downward corrected ULN of the former, as proposed by others [17], resulted in a slight improvement of sensitivity, at the cost of a decrease in specificity to the extent to become lower than that of both, U-MNs and U-CATs. We could not reproduce the small sensitivity of 74.1% of the EIAs, that has led others to suggest using a mathematical model for downward adjustments of the ULN. The lack of benefit of such an adjustment of ULN may have been due to the relatively small sample size of patients with PPGL, yet the number of PPGL in our study was identical to that of the German study [17]. Whether a larger number of PPGL would result in a significantly increased sensitivity remains to be determined.

The post-test probability (post-TP) that an individual is affected by a PPGL, given a positive or negative test result on the basis of the probability of the individual having the condition before the test was run (5.7%), was almost perfect and higher for the P-MNs (100% and 0%) than for U-CATs (96% and 1%), and for U-MNs (18% and 0%), respectively. Obviously, the ULN of the U-MNs (138 μg/24 h) and of U-NMN (311 μg/24 h) was too low, and has been changed by the manufacturer to 320 μg/24 h (U-MN) and to 390 μg/24 h (U-NMN) in February 2019, long after the study has been completed. Applying these higher ULN would have increased post-TP of PPGL given a positive test result from 18 to 76%, without changing the nearly perfect exclusion of PPGL. Given the potentially different results obtained with the new assay version after February 2019, this remains to be determined, however. Increasing ULN would increase specificity of the tests but be followed by an unacceptable decrease of sensitivity. Subjecting only patients with a high pre-test probability of PPGL to biochemical testing could lead to a better specificity. However, studies have not shown consistently better specificity (Appendix A) with such an approach, which in routine patient care does not seem practical either.

Initial proficiency studies have suggested that P-NMN measured by the EIA of LDN was systematically lower than determined by HPLC, even after correction for the D-enantiomer of spiked samples and was ascribed to matrix effects [28]. Further quality control studies have found that the linearity of the assay for P-MN exceeded the manufacturer’s claim of precision (that of P-NMN was in accord), and that both P-MN and P-NMN underestimated the Royal College of Pathologists Australasia Quality Assurance Program target value by 15.6% and 18.3%, respectively [29]. It was concluded, that the large negative bias (32%) of P-NMN measured by EIA versus by LC-MS/MS was not a matrix problem but instead related to the assay calibration [30]. Immunoassays target the natural L-enantiomer of P-MN, but calibration of the assay uses a racemic mixture of D- and L-enantiomers, because there is no commercially available calibrator material containing the L-enantiomer only [27]. Recently, measurements of P-MN by the EIA of LDN was 11% lower than those obtained by LC-MS/MS in a multicenter study of 341 patients resulting in a low sensitivity of 74.1% (14 of 54 PPGL false negative). Following the use of a recalculated lower ULN to correct for the negative bias of the EIAs, sensitivity increased to 96.2%, with only a small loss in specificity (99.3% to 95.1%). The authors conclude that until the calibration problem is corrected, ULN issued by the manufacturer should be adjusted to overcome this drawback [17]. We could not confirm these findings, suggesting that good patient care is possible using ULN stipulated by the manufacturers of the EIAs without corrections. It may be speculated that the manufacturer has resolved the calibration problem. LDN uses standards for the calibration produced from commercially available material containing racemic D-/L-MN and D-/L-NMN at a 1:1 ratio. For calibration of the assay, while the antibody used for MN detects the L-enantiomer only, that for NMN also detects the D- enantiomer at a rate of 80%. These binding properties are then used to calculate the sample weight for calibration: 1 mg MN = 0.5 mg and 1 mg NMN = 0.9 mg (personal communication, Klemm R).

The upright position leads to an increase of sympathetic output. As compared to the supine position, both plasma-free MNs (P-NMN more than P-MN) as measured by LC-MS/MS obtained from the patient in the sitting position have been suggested to result in an up to three-fold increase of false positives [31,32], so the 2014 Endocrine Society Clinical Practice Guidelines recommend that patients should be fully recumbent for at least 30 min before sampling [15]. One may assume a negligible influence of posture on the usually already highly elevated plasma concentrations of free metanephrines in patients with PPGL. However, drawing blood in the sitting position may be clinically important in those without the disease. The 25% increase of P-NMN [33] and the 27% and 30% increase of P-MN and P-NMN, respectively [31], as a result of change in posture, have been confirmed by some [34] but not all [35] authors. It has been acknowledged that blood sampling for P-MNs is frequently performed with the patient in the upright position during clinical routine work due to time constraints [36]. It is of note that ULN stipulated by the manufacturer of the EIAs of LDN were established from blood samples taken in the sitting position of 96 healthy volunteers; the first circumstance (non-recumbency) may result in falsely high and the latter (healthy individuals) in presumably falsely low ULN when results are to be interpreted in typical screening populations of hypertensive patients. Blood sampling in our study was in the supine position in about two-thirds before and in all patients after February 2013. This may have balanced out the systematic error potentially introduced, if one assumes falsely high ULN of P-MNs established from the non-recumbent position. The degree of deviation below the ULN of false negatives was similar among all pairs of analytes (Figure 3), reassuring that the benefit of the high sensitivity was not diminished by a large range of concentrations below the ULN of either metabolite. On the other hand, the range of false positives was higher with the combined 24 h-urinary compared to the free plasma MNs (to almost 5-fold vs. up to a maximum of 3.3-fold above the ULN, respectively), suggesting that positive findings of U-MNs are less reassuring than of P-MNs, but may have been the result of the obviously falsely low ULN of the former pair of metabolites.

In comparing P-MNs by LC-ED vs. the combination of U-MNs by spectrophotometry and of U-CATs by LC-ED, is has been suggested that P-MNs may be the test of choice in high-risk patients (with familial syndromes). However, in the more common clinical situation to screen for sporadic PPGL, especially in older hypertensive patients, measurement of U-MNs in combination with U-CATs may provide comparable sensitivity with a lower rate of false-positive tests. The authors acknowledge that defining the absence of PPGL by the clinicians´ evaluation or by an alternative diagnosis only was one of the limitations [37]. In our study, specificity of measurements of plasma-free MNs exceeded that of the 24 urinary-fractionated MNs and does not support their contention. Our findings are in line with some [17,38,39,40,41,42] but not all [30,43,44] authors assessing the performance of P-MNs measured by EIA, and is not related to sample size nor to pre-TP of PPGL of the study populations (Appendix A, respectively). Appendix A gives a summary of the diagnostic performance of the EIAs of the available literature for the quantification of plasma-free metanephrines.

PPGL may relapse at a rate of 1 in 100 patient years, or with a probability as high as 25% over 5 years in patients <20 years harboring PGL >15 cm at first diagnosis [45], and may metastasize even 15 years later in patients initially thought to harbor a benign tumor [46]. There are patients with biochemical results above the ULN, but without radiological evidence of pheochromocytoma/paraganglioma (PPGL). The clinical judgement here is either a false positive biochemical result of a person free of the tumor, or a correct positive due to a PPGL too small to be detected by even the most sensitive radiological method. In both cases, long-term follow-up (FU) appears to be mandatory. Still, studies not always provide such information (Appendix A). Given a large enough sample studied, it has been assumed, that a study´s potentially incorrect categorization of patients would not make a significant difference as to sensitivity and specificity, and that the conclusion of the study would not be different [47]. Patients may develop the disease during FU, and may not have suffered from PPGL initially, even if the first lab tests have been (mildly) positive. However, PPGL not discovered are potentially life-threatening [12,13,48], and suspicion of PPGL given a positive biochemical test result may always be prudent. Six of the pertinent studies assessing P-MNs by EIA in comparison to other methods do not mention a FU of their patients defined free of PPGL [30,40,42,43,44], or do not report the duration of FU [38]. Three other series provide clinical, or clinical and radiological FU for 1–2 years [17,38,41] which is in line with our approach (Appendix A). One way of interpreting concentrations above the ULN could be to not categorize test results as either “positive” or “negative”, but instead as a spectrum from “PPGL likely” to “PPGL unlikely”. It may be remembered in that context, that lab reference intervals usually contain 95% of the healthy population. Thus, 2.5% of healthy persons by definition have concentrations of biochemical parameters slightly above the ULN. Presenting as patients, it may not be possible to differentiate between those with and without PPGL. However, imaging and the clinical impression may help to decide, whether a positive test result is in fact ill-advised or not [48,49]. This “clinical reasoning” [50] may—in addition to the decreasing patient adherence to FU over time—provide an explanation why in clinical practice not every “positive” finding results in further diagnostic procedures. Sufficiently large study samples may be the best way to avoid bias, however [51]. The larger the study population, the smaller the 95% CI of reported sensitivities and specificities, the more credible the conclusions drawn. This applies to our study and to all series with patients considered free of the disease by far exceeding those with PPGL (Appendix A). Only a small fraction (8%) of patients with ≥1 positive measurements of any parameter had surgical confirmation of diagnosis (Figure 1). However, our study population was the largest of the literature investigating the performance of plasma free metanephrines by EIA for the diagnosis of PPGL (Appendix A), and there is only one study [17] presenting data of an equal number of patients with patho-histological confirmation of diagnosis. The latter was a multicenter study, while our patients were all from a single hospital.

There are several limitations to our study. We have performed a retrospective analysis with a limited number of patients with PPGL, which may have compromised power to detect sensitivity difference. However, the overall prevalence of PPGL is very low, and a prospective study would need a very large sample population and is not available as yet. In our study, all patients were well characterized, and the sample size was among the largest of the studies that assessed the performance of an EIA for P-MNs. Also, our study was conducted in one center, potentially limiting generalizability, although it is of note that we presented data of a commercially available EIA kit widely used. We did not measure 3-methoxytyramine, the Catechol-O-Methyltransferase metabolite of dopamine, the “sibling” of MN and NMN, that recently has become available and included in the new kits of the EIA. Thus, purely dopamine-secreting PPGLs could not be detected. Moreover, drug history was not available, so specificity may have been overestimated. Interference by medications has been studied for measurement of P-MNs by LC-MS/MS, but there are no data with EIA. Incomplete collection of 24 h-urine could have resulted in falsely low U-MNs and U-CATs measurements. The use of an inappropriately low ULN of 24 h-urinary fractionated metanephrines may have compromised routine patient care, unless other diagnostic criteria have been considered in the treatment decisions. With use of the increased ULN issued by the manufacturer after February 2019, post-TP of PPGL given a positive test of the combined U-MNs improved but was still significantly below that of the pair of P-MNs. Also, we did not study P-MNs by HPLC-MS/MS, so no inference can be drawn as to that method.

One of the strengths of our study is that samples were provided by the central lab of the largest hospital in our country serving a population of more than 2 million. Additionally, all three pairs of metabolites (P-MNs, U-MNs, U-CATs) were available in every patient and at the same time within a broad spectrum of almost 1000 well-characterized patients suffering from refractory hypertension, anxiety disorders, or adrenal incidentaloma, which may be confused with PPGL, rendering the study sample representative of a typical population treated by endocrinologists including hereditary syndromes. Finally, for the first time, data were compared with use of a recently proposed ULN to adjust for the assumed systematic negative bias of the EIAs for P-MNs in addition to that of ULN stipulated by the manufacturer [17].

## 5. Conclusions

In conclusion, we have shown that the determination of P-MNs with a commercially available EIA kit is as sensitive and more specific when compared to U-MNs by HPLC, and that both methods perform better than U-CATs, the latter resulting in a low sensitivity despite excellent specificity. In the absence of HPLC-MS/MS available, state-of-the-art care of patients suspected to harbor PPGL may thus be possible with the EIAs studied. Post-test probabilities of PPGL given a positive and negative test result of P-MNs were almost 100% and 0%, respectively, with P-MNs, but only 18% and 0% with U-MNs. The systematic negative bias of P-MNs resulting in a low sensitivity by EIAs as reported previously [17] could not be reproduced, suggesting that the downward correction of ULN, as proposed by others, when measuring P-MNs b EIA [17], are not essential.

## Figures and Tables

**Figure 1 jcm-09-03108-f001:**
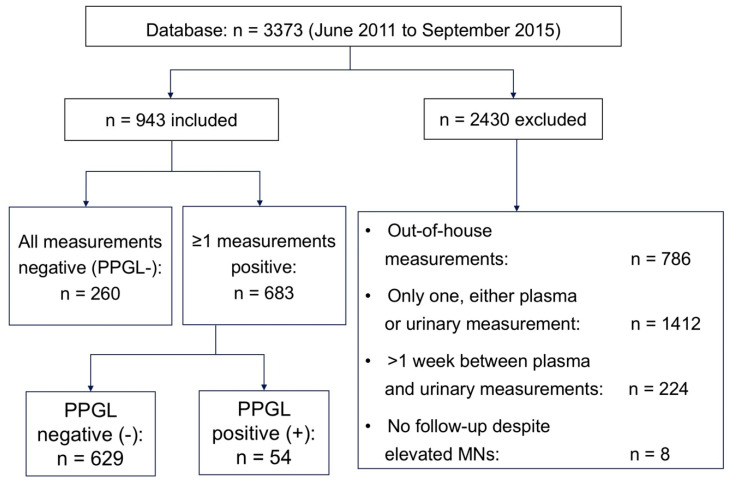
Flow-chart of patients.

**Figure 2 jcm-09-03108-f002:**
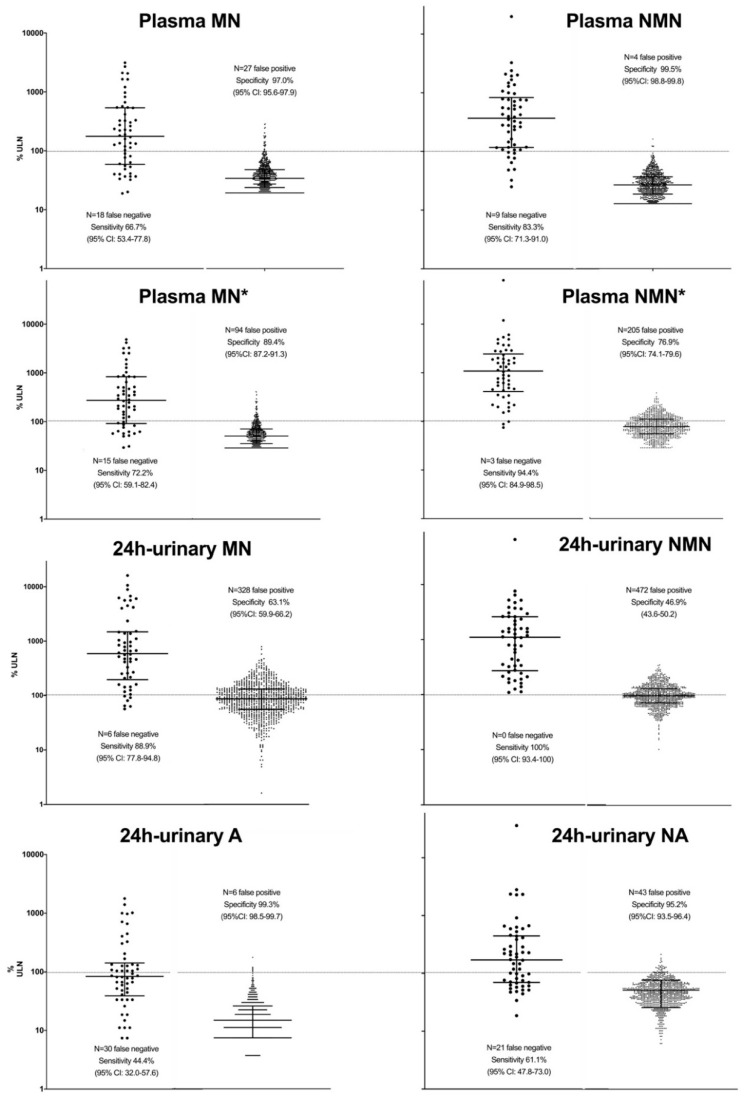
Sensitivities and specificities of single parameters with all ULN. Data are presented for 54 patients (left columns), and for 889 individuals without (right columns) PPGL, respectively. Concentrations are expressed as percentages of the respective ULN (dashed line). Lines within the graph depict medians and inter-quartile ranges. The asterisks (*) denote P-MN and P-NMN with the corrected ULN as proposed recently [17]. Please note the logarithmic scale!

**Figure 3 jcm-09-03108-f003:**
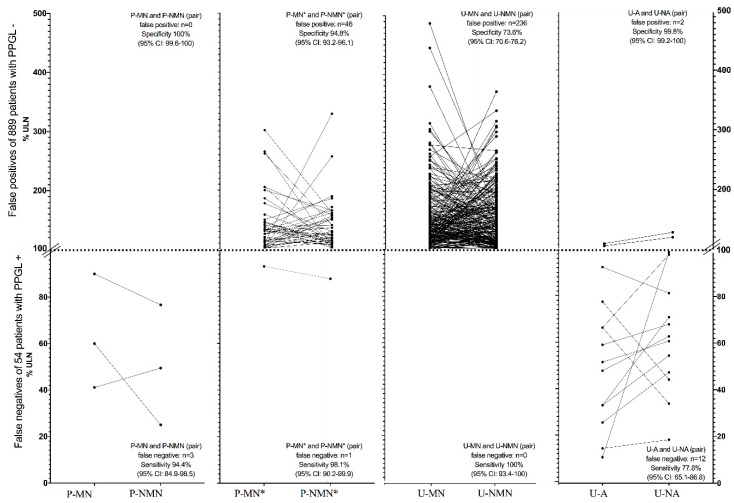
False negative results of 54 patients with PPGL+ (below dotted line depicting the ULN) and false positive results of 889 patients with PPGL- (above dotted line) of the combined parameters (P-MN+P-NMN, U-MN+UNMN, U-A+U-NA). Intraindividual measurements within the pairs of parameters are connected with solid lines. The asterisks (*) denote P-MN and P-NMN with the corrected ULN as proposed by others [17]. Please note the different scale below and above the dotted line.

**Figure 4 jcm-09-03108-f004:**
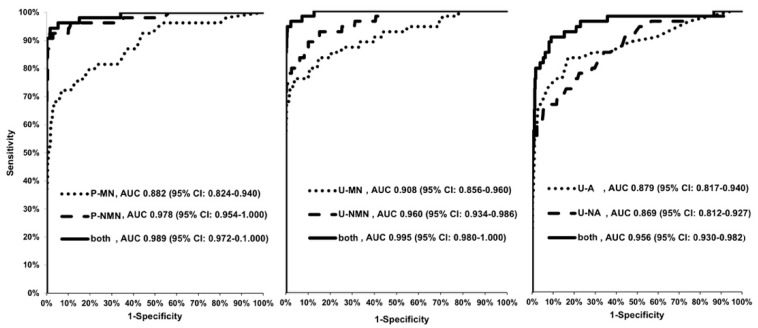
Area under the curve (AUC) of receiver–operator curve (ROC) of single parameters and of the combined model (pairs of metabolites).

**Table 1 jcm-09-03108-t001:** Upper limits of normal (ULN) of the metabolites as used in the analyses of this study.

Biochemical Parameter	ULN
P-MN	90 pg/mL
P-NMN	180 pg/mL
P-MN *	58 pg/mL
P-NMN *	45–88 pg/mL
U-MN	138 µg/24 h
U-NMN	311 µg/24 h
U-A	27 µg/24 h
U-NA	97 µg/24 h

The asterisks (*) denote P-MN and P-NMN with the corrected ULN as proposed by others [17]: In short, the two values for P-NMN * are the lowest and highest concentrations of the age-adjusted ULN, from 45 pg/mL at the age of 5 years to a maximum at the age of 65 years and above. Details are given in the legend of Table 3 of their work [17]. To convert values for plasma measurements to picomoles per liter, and for urinary measurements to nmol/day, multiply by 5.46 for normetanephrine, 5.08 for metanephrine, 5.91 for norepinephrine, and 5.46 for epinephrine. Please note that after the study has been completed, the lab has issued new ULN for P-MN and P-NMN (68 and 196 pg/mL, respectively), and for U-MN and U-NMN (311 and 390 µg/24 h, respectively), but patient care and all analyses were by the above ULN, respectively.

**Table 2 jcm-09-03108-t002:** Patients with at least one parameter above the lowest respective upper limits of normal (ULN) (see Table 2) and allocated to correct positive (PPGL+) and correct negative (PPGL-) by best possible criteria as outlined in the text.

*n*	PPGL+	N	PPGL−
52	histology+	2	histology−
2	lab+ and J- MIBG-scintigraphy+ (1) and F-DOPA PET-CT+ (1)	158	lab and radiol FU−
		190	lab and clinical FU−
		10	alternative diagnosis
		269	clinical FU−

Please note, that the *n* = 260 patients with all parameters below the respective ULN (for P-MN and P-NMN: Both had to be below the respective downward corrected ULN [17]), add to the total number of PPGL- patients.

**Table 3 jcm-09-03108-t003:** Characteristics of the study population of *n* = 943 patients and concentrations of biochemical parameters (median and absolute range).

Patients	*n* = 54 PPGL+	*n* = 889 PPGL−	*p*-Value
Age (years)	51 (18–76)	55 (5–90)	0.55
Female sex, *n* (%)	28 (52)	537 (60)	0.80
Genetic syndrome, *n* (%)	14 (26)	0	
P-MN (pg/mL)	159 (17–2815)	30 (17–248)	0.001
P-NMN (pg/mL)	663 (45–35,832)	48 (23–290)	0.001
U-MN (μg/24 h)	814 (78–22,248)	118 (5–672)	0.001
U-NMN (μg/24 h)	3426 (340–203,009)	321 (33–1388)	0.001
U-A (μg/24 h)	23 (2–480)	4 (1–47)	0.001
U-NA (μg/24 h)	166 (18–34,987)	46 (6–314)	0.001

Please note the LLOQ of 17 and 23 pg/mL for plasma-NMN and -MN, and of 33, 5, 6, and 1 μg/24 h for 24 h-urinary NMN, -MN, -NA, and -A, respectively. To convert values for plasma measurements to picomoles per liter, and for urinary measurements to nmol/day, multiply by 5.46 for normetanephrine, 5.08 for metanephrine, 5.91 for noradrenaline, and 5.46 for adrenaline, respectively.

**Table 4 jcm-09-03108-t004:** Sensitivity (above) and specificity (below) of the pair of plasma compared to the pair of 24 h-urinary metabolites, respectively.

**Comparison EIA vs. HPLC**	**EIA (Plasma) Sensitivity (95% CI); ULN**	**HPLC (24 h-Urine) Sensitivity (95% CI); ULN**	***p*** **-Value**
P-MN + P-NMN/U-MN + U-NMN	94.4% (84.9–98.5); 90 and 180 pg/ml	100% (93.4–100); 138 and 311 μg/24 h	0.25
**P-MN + P-NMN**/U-A + U-NA	94.4% (84.9–98.5); 90 and 180 pg/mL	77.8% (65.1–86.8); 27 and 97 μg/24 h	**0.02**
P-MN * + P-NMN */U-MN + U-NMN	98.1% (90.2–99.9); 58 and 45-88 pg/mL	100% (93.4–100); 138 and 311 μg/24 h	0.88
**P-MN * + P-NMN ***/U-A + U-NA	98.1% (90.2–99.9); 58 and 45-88 pg/mL	77.8% (65.1–86.8); 27 and 97 μg/24 h	**0.008**
**Comparison EIA vs. HPLC**	**EIA (Plasma) Specificity (95% CI); ULN**	**HPLC (24 h-Urine) Specificity (95% CI); ULN**	***p*** **-Value**
**P-MN + P-NMN**/U-MN + U-NMN	100% (99.6–100); 90 and 180 pg/mL	73.6% (70.6–76.2); 138 and 311 μg/24 h	**0.001**
P-MN + P-NMN/U-A + U-NA	100% (99.6–100); 90 and 180 pg/mL	99.8% (99.2–100); 27 and 97 μg/24 h	0.36
**P-MN *+ P-NMN ***/U-MN + U-NMN	94.8% (93.2–96.1); 58 and 45–88 pg/mL	73.6% (70.6–76.2); 138 and 311 μg/24 h	**0.004**
P-MN * + P-NMN */**U-A + U-NA**	94.8% (93.2–96.1); 58 and 45–88 pg/mL	99.8% (99.2–100); 27 and 97 μg/24 h	**0.01**

Higher sensitivities and specificities are depicted in bold. The asterisks denote the corrected ULN [17].

**Table 5 jcm-09-03108-t005:** Comparisons of the AUC of ROC curves (of single parameters above, of combined parameters below).

**EIA (Plasma)/HPLC (24 h-Urine)**	**EIA (Plasma) AUC (95% CI)**	**HPLC (24 h-Urine) AUC (95% CI)**	**z-Value**	***p*** **-Value**
P-MN/U-MN	0.882 (0.824–0.940)	0.908 (0.856–0.960)	0.64	0.52
P-MN/U-NMN	0.882 (0.824–0.940)	**0.960 (0.934–0.986)**	2.39	**0.02**
P-MN/U-A	0.882 (0.824–0.940)	0.879 (0.817–0.940)	0.07	0.94
P-MN/U-NA	0.882 (0.824–0.940)	0.869 (0.812–0.927)	0.31	0.76
P-NMN/U-NMN	0.978 (0.954–1.000)	0.960 (0.934–0.986)	1.02	0.31
P-NMN/U-MN	**0.978 (0.954–1.000)**	0.908 (0.856–0.960)	2.37	**0.02**
P-NMN/U-A	**0.978 (0.954–1.000)**	0.879 (0.817–0.940)	2.90	**0.004**
P-NMN/U-NA	**0.978 (0.954–1.000)**	0.869 (0.812–0.927)	3.47	**<0.0001**
**EIA (Plasma)/HPLC (24 h-Urine)**	**EIA (Plasma)AUC (95% CI)**	**HPLC (24 h-Urine) AUC (95% CI)**	**z-Value**	***p*** **-Value**
P-MNs/U-MNs	0.989 (0.972–1.000)	0.995 (0.980–1.000)	1.02	0.31
P-MNs/U-CATs	**0.989 (0.972–1.000)**	0.956 (0.930–0.975)	1.97	**0.04**
U-MNs/U-CATs	**0.995 (0.980–1.000)**	0.956 (0.930–0.982)	2.12	**0.03**

Higher AUCs are depicted in bold.

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
