# Peer review of "Measurements of Plasma-Free Metanephrines by Immunoassay Versus Urinary Metanephrines and Catecholamines by Liquid Chromatography with Amperometric Detection for the Diagnosis of Pheochromocytoma/Paraganglioma"

_jcm, 2020, doi:10.3390/jcm9103108_

Round 1

Reviewer 1 Report

The authors have responded  satisfactorily to the minor issues, however I do not agree about the fact that HPLCMS/MS is, "error-prone, difficult to calibrate, and rarely available". This method is  used in many countries and has been extensively described in many publications within the last 5-10 years. Only in the RCPA program (Royal College of Pathologists of Australasia) of external quality control, 50 laboratories sent their metanephrine results to this center. Metanephrine quantification represents the gold standart for PHEO/PGL even though it is expensive is has been shown to be the best diagnosis methods for these kind of tumors. I am very surprised that no laboratories offers this kind of quantification in Austria (being a rich country) but most developped countries have implemented this technology either in private laboratory or in university hospital.  To note there are 120 articles in medline (as of 25.8.2020) retrieved with the key words "metanephrines and mass spectrometry" within the last 4 decades or more. I fully understand that in the absence of HPLCMS/MS it is of interest to compare which is the best methods for which metabolites to be analyzed and comparing these methods between each other. Following this approach the paper would be interesting for analysts from countries where HPLC-MS/MS is not available.

Author Response

Thank you very much for your positive comments. Please find our answers below.

 Reviewer:

The authors have responded satisfactorily to the minor issues, however I do not agree about the fact that HPLCMS/MS is, "error-prone, difficult to calibrate, and arely available"…. represents the gold standart for PHEO/PGL even though it is expensive…

We agree, that HPLC-MS/MS are among the first-line tests for quantification of P-MNs. We concede, that refinements of the methods have resulted in easier handling, and lower costs. Please note, that we have eliminated the mentioned statement (lines 317 ff. of the first version of the manuscript). We have instead re-phrased parts of the introduction (also following the suggestions of reviewer 4), containing the wording of the publication of Därr et al., introduced as new reference [17]: “Capital costs of instrumentation, the level of operator expertise, and the need for in-house validation are higher than with EIA methods.” (lines 64 ff.).

Please note, that this publication has been added as a new reference [17], and the successive ones adapted accordingly. In addition, we apologize, that reference [46] was wrong and had to be replaced.

It may be important to again stress the fact, that HPLC-MS/MS is still not available in Austria, a fact, that has been more clearly indicated in the Introduction as one of the reasons why the study was undertaken (lines 61 ff.). Said that, thank you for your understanding, that you consider the paper interesting for analysts from countries where HPLC-MS/MS methods are not available as well.

Please note, that the other changes in the manuscript are the consequence to the comments of reviewer 4.

We hope, that you will find the revised paper now suitable for publication in the Journal of Clinical Medicine.

Sincerely yours.

The authors.

Reviewer 2 Report

The authors have satisfactorily addressed the reviewers comments and improved the manuscript. No further comments.

Author Response

Thank you very much for reviewing and helping to improve our manuscript.

Reviewer:

The authors have satisfactorily addressed the reviewers comments and improved the manuscript. No further comments.

Thank you very much!

The authors

Reviewer 3 Report

The authors have made a reasonable attempt at responding to the questions and comments raised earlier. 

I have no additional concerns.

Author Response

Thank you for reviewing and helping to improve our manuscript.

Reviewer:

The authors have made a reasonable attempt at responding to the questions and comments raised earlier. I have no additional concerns.

Thank you very much!

The authors

Reviewer 4 Report

The manuscript by Raber et al., aims to analyses the difference in specificity and sensitivity of two different methods for the measurements of catecholamines in patients affected by pheochromocytoma/paraganglioma. In particular, by mean of a retrospective analysis they compare the measurements of plasma free matanephrines by immunoassay versus urinary metanephrines by liquid chromatograpy with amperometric detection. The study is complex and well organized, the methods and the statistics adequately described, however the results obtained need to be more accurately described and presented, and more importantly, the clinical relevance of the results needs to be emphasized by the authors. Finally, the discussion is too long and does not critically analyses to the significance of the results obtained.

Major points:

  1. Introduction: the aims of the study are not clearly explained: what is the reason why the authors performed this study? What are the limitations related to the diagnostic methods routinely used for pheochromocytoma/paraganglioma and what the improvements coming from their study?
  2. Results section: the data reported in figure 2, 3 and 4 needs to be explained in a more exhaustive manner. What is the importance to normalize the plasma measurements with the urinary detection of catecholamines? How the normalization is reported in the graphs?
  3. Discussion: needs to be finalized to the critical evaluation of the results obtained describing them straightforward, underlining their value and their usefulness from a diagnostic point of view in terms of specificity and sensitivity of the methods investigated. Finally, indicate how the bias of the methods analyzed might be overcame in order to improve the specificity of the tests.
  4. Check the English for the typos 

All these aspect of the study needs to be clarified in order to improve the quality of the paper and to allow their publication on J of Clinical Medicine.

Author Response

Reviewer 4:

Thank you very much for your helpful comments. Please find our answers to your points below.

Reviewer:

  1. Introduction: the aims of the study are not clearly explained: what is the reason why the authors performed this study? What are the limitations related to the diagnostic methods routinely used for pheochromocytoma/paraganglioma and what the improvements coming from their study?

The third paragraph of the introduction (lines 61 ff.) has been changed to the following to more clearly explain:

“To date, no laboratory of the health care system of the Republic of Austria, in 2020 serving an estimated population of 8.9 millions (https://en.wikipedia.org/wiki/Austria), has been offering chromatographic/MS methods for quantification of plasma free metanephrines for routine patient care, neither in private institutions nor in university hospitals. Capital costs of instrumentation, the level of operator expertise, and the need for in-house validation are higher than with EIA methods [17]. It is of note, that in the European Union, until only recently, there has been no single officially CE-IVD (Conformité Européenne, In-Vitro Diagnostic Medical Device) approved mass spectrometric or coulometric detector for the high-pressure (HP)LC to quantify plasma free metanephrines - however, this approvement is mandatory for many labs due to regulatory legal requirements for the implementation of new methods. Based on the non-availability of these methods, we report our experience of n=943 patients tested for PPGL in routine clinical care, in every patient and simultaneously using the three test methods routinely available (plasma free metanephrines by EIA, 24-hour urinary fractionated metanephrines, and 24-hour urinary catecholamines, the latter two by HPLC with amperometric detection), compute which performs best for which metabolites to be analyzed, and compare the methods between each other. We think, that the question, whether state-of-the-art care is possible without HPLC-MS/MS, is of interest for analysts in all countries, where HPLC-MS/MS is not available. In 2015, the commercially available EIAs of Labor Diagnostika Nord, Nordhorn, Germany (LDN) have been reported to systematically underestimate plasma free metanephrines contributing to a poor diagnostic sensitivity of only 74.1% in one series [16]. We thus wanted to know, whether the correction algorithm, as proposed by Weismann et al. [16] to overcome their published negative bias of the EIAs suggested to have caused the low diagnostic sensitivity [16], is of clinical importance in a different population, or whether patient care would be possible without using these downward corrected upper limits of normal (ULN) of plasma free metanephrines.”

Please note, that a new reference [17] has been added to that paragraph, and the successive ones adapted accordingly. In addition, we apologize, that reference [46] was wrong and had to be replaced.

Reviewer

  1. Results section: the data reported in figure 2, 3 and 4 needs to be explained in a more exhaustive manner.

The data reported in Figures one (lines 252 ff), two (lines 269 ff.) and three (lines 308 ff.) have been presented more extensively. Please note, that Figure 4 has been corrected as to the value of the AUC of U-NMN.

What is the importance to normalize the plasma measurements with the urinary detection of catecholamines? How the normalization is reported in the graphs?

Concentrations given in the literature vary, sometimes given in traditional units, sometimes according to the Système International (SI). To avoid for confusion because of different units for plasma free metanephrines (in our lab issued as pg/ml) and for 24-hour urinary metanephrines and -catecholamines (in our lab issued as μg/24 hours), and to allow for easy visual comparison between the analytes, presentation of the data in Figures two and three was by normalization to the upper limit of normal (ULN). As explained in the respective legends, this was done by presenting concentrations as percentages of the respective ULN (depicted by a dashed line).

In addition, part of the legend of Table 2 (lines 188 ff.) has been re-written as follows to more clearly explain, that the corrected ULN of plasma free metanephrines were proposed by others, and how they were computed:

“The asterisks (*) denote P-MN and P-NMN with the corrected ULN as proposed by others [16]: in short, the two values for P-NMN* are the lowest and highest concentrations of the age-adjusted ULN, from 45 pg/ml at the age of 5 years to a maximum at the age of 65 years and above. Details are given in the legend of Table 3 of their work [16]….”

Reviewer

  1. Discussion: needs to be finalized to the critical evaluation of the results obtained describing them straightforward, underlining their value and their usefulness from a diagnostic point of view in terms of specificity and sensitivity of the methods investigated.

The discussion has been diligently shortened, re-organized and finalized as to a more concise and straightforward description of the clinical impacts of our study. Please note, that the former third paragraph of the discussion (starting with “To the best of our knowledge…”) has been moved to the beginning of the discussion. All the changes have been carefully marked.

Finally, indicate how the bias of the methods analyzed might be overcame in order to improve the specificity of the tests.

We did not compare concentrations of MNs in plasma versus urine, we compared the performance of the tests for the diagnosis of PPGL. We found a high sensitivity of P-MNs by EIA (only 3 of 54 [5.5%] PPGL missed), not different to that of U-MNs, which are among the first line tests recommended by the Endocrine Society [15]. We thus concluded, that there is no bias as suggested by others [16]. This has been discussed extensively in the revised text. Please note, that the word “bias” has been changed to “systematic error” in line 523 to avoid confusion with the bias suggested by others [16] for the measurement of P-MNs by EIA.

The following has been introduced (lines 403 ff.) as to possible improvement of specificity:

“Increasing ULN would increase specificity of the tests, but would be followed by an unacceptable decrease of sensitivity. Subjecting only patients with a high pre-test probability of PPGL to biochemical testing could lead to a better specificity. But studies have not shown consistently better specificity (Supplementary Figures 2 and 3) with such an approach, which in routine patient care does not seem practical either.”

Please note, that the numbering of the Supplementary Figures have changed according to their appearance in the text.

Reviewer:

  1. Check the English for the typos

Thank you. We checked the typos carefully, and will ask the Journal to double-check.
